# Cardiovascular disease risk among Australian unpaid carers – A survival analysis using 15 waves of the HILDA survey

Ameer Lambrias[‡], Yamna Taouk[‡], Jennifer Ervin, Tania King*

Centre for Health Policy, Melbourne School of Population and Global Health, University of Melbourne, Melbourne, Australia

‡ These authors share first authorship on this work.
* tking@unimelb.edu.au

## Abstract

### Background

Engaging in chronically stressful behaviours has been hypothesised to increase the risk of experiencing cardiovascular disease (CVD). Providing unpaid care is known to be a stressful activity, but it is not clear whether this caregiving is associated with CVD. This study filled a gap in the existing literature by examining the association between providing unpaid care and incident cardiovascular disease among a nationally representative sample of Australian adults.

### Methods

11,123 adult participants aged over 18 years from the Household Income and Labour Dynamics in Australia (HILDA) survey were followed for up to 14 years from baseline (2003) until 2017. Gender-stratified survival analysis models used self-reported caregiving and heart disease statuses as well as time-varying covariates, to assess the association between providing high-intensity or low-intensity unpaid care (to an elderly or disabled relative) and incident CVD in comparison with a non-caregiving control.

### Results

Among females, there was weak evidence that CVD was associated with high-intensity unpaid care (HR = 1.27, 95% CI = [0.83, 1.95]) and no evidence for low-intensity unpaid care (HR = 0.79, 95% CI = [0.50, 1.26]) in comparison with non-carers after adjusting for confounders. There was no association between caregiving and incident CVD for high-intensity (HR = 0.82, 95% CI = [0.47, 1.42]) or low-intensity (HR = 0.84, 95% CI = [0.55, 1.28]) caregiving males in the adjusted models.

**Data availability statement:** The data used for this study were collected by the Melbourne Institute of Applied Economic and Social Research. Applications for data access is through the Australian Government Department of Social Services Longitudinal Studies Dataverse (https://dataverse.ada.edu.au/dataverse/DSSLongitudinalStudies). There are some restrictions on accessing this data, and it is not available to the public. Citation: Australian Government Department of Social Services. 2023. ADA Dataverse. DSS Longitudinal Studies Dataverse. The Household, Income and Labour Dynamics in Australia (HILDA) Survey. https://www.dss.gov.au/about-the-department/longitudinal-studies/living-in-australia-hilda-household-income-and-labour-dynamics-in-australia-overview

**Funding:** This study was supported by Australian Research Council grants (DE200100607, LP180100035 & DP250101371 to TK), and the University of Melbourne (Dame Kate Campbell Fellowship to TK).

**Competing interests:** The authors have declared that no competing interests exist.

## Conclusions

These findings do not provide strong evidence to reject the null hypothesis that providing unpaid care does not increase risk of developing CVD in the Australian population. Given that these findings are somewhat inconsistent with the extant literature from other populations, further research is necessary, both in Australia and internationally, to build on the findings of this study and improve understanding of the nature of the association between caregiving and incident CVD.

## Introduction

Unpaid care is a vital social resource to aid in fulfilling the care needs of vulnerable people. It also serves to reduce demands and cost burdens placed upon formal health institutions. The economic value of the care performed by unpaid carers in Australia has been estimated to be equivalent to AU$78 billion annually, representing a value that is equal to 4% of the country's total GDP [1]. Globally, it has been estimated that there are 16.4 billion hours of unpaid care work performed each day [2]. In comparison with population groups who do not provide care to others, unpaid carers have been observed to experience reduced time for leisure and self-care activities [3]. Additionally, there are increased financial costs that are incurred when supporting the needs of a person who cannot independently sustain themselves [4]. Moreover, the demands of providing unpaid care to others can significantly reduce the availability and capacity of carers to be able to fulfil their occupational responsibilities [5].

Particular social determinants of health can affect the likelihood of an individual performing unpaid care work, including age, gender, and socioeconomic status [6]. In Australia, the likelihood of providing unpaid care is positively correlated with age throughout the population until the 55–64 year old age group, with reductions seen in the over-65 population [1]. However, the number of unpaid carers aged 65–74 or 75 + has increased between 2015 and 2020 [1]. It is also estimated that 11% of the population (2.7 million people) act as unpaid carers to others [1], with Australian women contributing a majority of the nation's unpaid care work - equivalent to 57.3% of the national total [7]. Additionally, primary caregiving prevalence is greater among Australians who fall into the two lowest quintiles of weekly income, whereas the national likelihood of being a non-carer is positively correlated with increases in income across these quintiles [1].

Unlike formal care workers, unpaid carers typically have an existing relationship with their care recipient [7]. Among individuals who provide care to a loved one, a 'family effect' of caregiving has been observed [8]. This family effect suggests that the wellbeing of the family carer is contingent on the health and welfare of their care recipient [9], such that the stress produced from acting as a caregiver can be exacerbated depending on the condition of their kin care recipient. Research which examines the family effect has reported that family carers suffer adverse mental health effects as the health of their parents or spouses deteriorated [10,11]. The relationship quality between the unpaid carer and their care recipient can also positively influence

the wellbeing of both individuals. An integrative review found that having higher quality relationships provided benefits for both unpaid carers and their care recipients, observing decreases in measures of distress and caregiving burden among carers who were considered to have high quality relationships with their care recipients [12].

Unpaid care has been associated with increases in cardiovascular disease (CVD) risk factors, suggesting that performing this unpaid work can confer negative health effects. Risk factors including hypertension [13], depressive symptoms [14], and increases in Framingham CVD risk scores, which predict the likelihood of experiencing a heart attack in the next decade [15], have all been linked with unpaid caregiving. However, some studies have observed reduced mortality among caregivers versus a non-caring control group [16–18], suggesting that unpaid care may be observed to produce positive health effects. The relationship between unpaid caring and specific CVD outcomes is less well defined. The association between providing unpaid care and cases of incident CVD was first observed in a study by Vitaliano et al. [19] who reported that participants who engaged in chronically stressful behaviours, such as providing care to their spouse, were at increased risk of developing coronary heart disease (CHD).

A recent systematic review examined the incidence of CVD amongst unpaid carers compared to non-caregiving controls [20], and found evidence that the risk of developing CVD among unpaid carers was modified by the intensity of the care provided by the caregiver. Prior studies which have categorised the measurement of the caregiving exposure into high-intensity and low-intensity groups have observed an association between providing intense levels of unpaid care and CVD [21–23]. In contrast, other studies which measured the caregiving exposure as a binary variable have yielded more mixed results. Some studies found no association between providing care to others and increased risk of developing CVD outcomes [24–26], whereas other studies observed increases in the incidence of CVD outcomes amongst family caregivers compared with a control group [27,28]. Additionally, a Northern Irish study identified a protective association between providing care to an ill, disabled, or elderly person and CVD-related mortality, with more intense caregiving (≥20 hours per week) associated with lower risk of mortality [17]. Moreover, whilst prior studies have explored the association between unpaid caring and CVD outcomes in different populations and settings, none have been undertaken in the Australian population. Furthermore, scant studies have accounted for the intensity of caregiving provided while exploring how this might influence any observed association.

It is noteworthy that the distributions of both unpaid caregiving and CVD in the Australian population are gendered. As previously mentioned, the unpaid care contribution of women exceeds that contributed by men [7]. In contrast, CVD is higher amongst men compared to women [29]. In Australia, 1.2 million people aged 18 and over (6.2% of the adult population) were living with one or more conditions related to heart, stroke or vascular disease in 2017–18 [29]. Based on self-reported data, the prevalence of heart, stroke and vascular disease among adults was higher among men (641,000, an age-standardised rate of 6.5%) than women (509,000, an age-standardised rate of 4.8%) [29]. However, there are key distinctions in the development of CVD by gender in Australia. Whilst CVD is more prevalent in males (compared to females) over the age 45, the prevalence of CVD among females is reportedly twice that of males in the under-45 population [29]. Gender differences in the severity of CVD can also be observed. For example, women are more likely than men to be hospitalised or die due to hypertensive disease, with atrial fibrillation and stroke both also causing greater mortality amongst Australian women than men [29].

Addressing key gaps in the extant literature, this study aims to: examine associations between providing unpaid care and the incident of CVD among Australian adults; and assess whether this association varies by the intensity of care provided, as well as by gender.

## Methods

### Study population

Data from the Household, Income and Labour Dynamics in Australia (HILDA) survey was used. Commencing in 2001, HILDA is Australia's first nationally representative longitudinal survey, amassing over 20 annual waves of data since it

began [30]. The survey follows over 17,000 participants each year, collecting information about their household and family relationships, income, employment, health, and education [31]. Data is collected annually from the same households and participants, using a combination of participant interviews as well as a self-completion questionnaire [32]. The first wave of the HILDA survey included 13,969 respondents residing in 7,682 different households, retaining the majority of this cohort in subsequent years. The number of participants included in the HILDA survey was allowed to fluctuate as household members moved in and out of participating residences, or existing family members became eligible for inclusion (e.g., by turning 15 years old).

## Study sample

This analysis was restricted to adult participants (18 years and over) who did not have CVD at the start of follow-up (2003). Self-reported CVD status was first measured in wave 3 (2003) of the HILDA survey and thus was selected as the baseline time at which participants must be active in the study. Follow-up continued until wave 17 (2017), the most recent wave at which CVD status had been measured. There were 12,822 participants aged over 18 years at baseline in wave 3 of the HILDA survey. Of these, 557 (4.3%) were excluded from the analysis, due to positive indication of CVD at baseline. A further 1,010 individuals (8.2%) were excluded as their caring intensity (exposure) status at baseline could not be ascertained. Finally, 132 participants (1.2%) were removed as they had missing data for covariates included in the analyses. This resulted in a final analytic sample size of 11,123 participants, comprising 5,927 females and 5,196 males (see Fig 1).

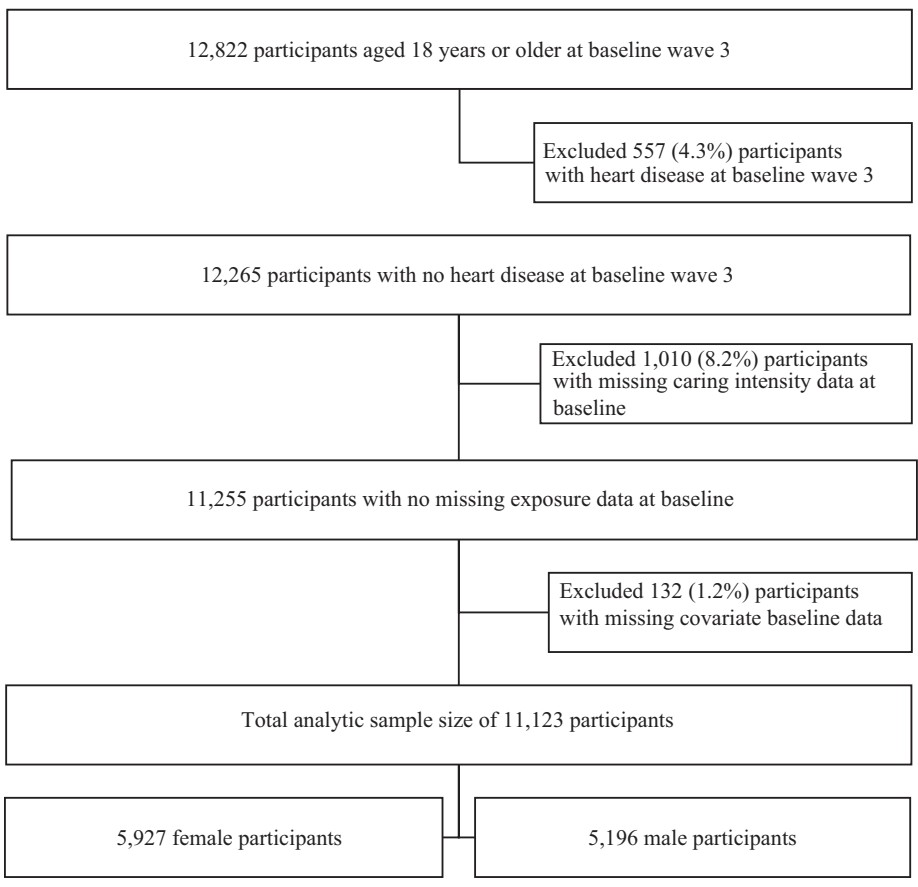

**Fig 1. Flow chart of sample selection for HILDA survey participants.**

## Unpaid care

Unpaid care was measured in each wave of the HILDA survey from wave 2 onwards as the combined total hours or minutes per week spent providing care to an elderly or disabled relative. Responses were self-reported using the self-complete questionnaire and coded in the HILDA dataset as a continuous numerical variable. To enable comparisons between high- and low-intensity care provision in this analysis, the data was classified into a three-tiered categorical variable comprising of: 0 hours of care per week (no care), >0 and <14 hours of care per week (low-intensity care), and ≥14 hours of care per week (high-intensity care). The 14 hour per week threshold of high-intensity caregiving was informed by prior research on caregiver health utilising this cut-off point [21,33,34].

## Incident CVD

Two variables were used to report heart disease status in HILDA, and these were used to derive the measure of incident CVD. The first variable used asked participants in waves 3, 7 and 9 to self-report whether they had heart disease as part of the self-completion questionnaire, noting that only participants who had reported having a long-term health condition were asked this question. A second heart disease variable was introduced in wave 9, replacing the first one. The second heart disease variable was administered as part of the interview, and asked participants who had reported that they had a long-term health condition if they had ever been told by a doctor or nurse that they had heart disease. The second variable was again measured in waves 13 and 17. In this analysis, these two variables were combined into a binary yes/no variable for CVD outcome status, using the first HILDA variable for waves 3, 7 and 9, and the second HILDA variable for waves 13 and 17. As such, our outcome (incident CVD) was able to be measured at waves 3, 7, 9, 13, and 17. We regarded the onset year of CVD as the year when participants first reported a new CVD diagnosis following baseline. As it was only participants with a long-term health condition that were asked to report their heart disease status in the HILDA data collection process, it was considered that individuals who reported having no long-term health conditions in this analysis also did not have CVD.

## Covariates

Potential confounders were identified a priori based on the existing literature These were age, gender, Indigenous ethnicity, country of birth, education, household structure, long-term health condition, labour force status, location, and household income. Aside from variables that were not expected to change over time (gender, Indigenous ethnicity, country of birth), all covariates were treated as time-varying and were assessed for all participants in each wave. Country of birth was grouped as Australia, other English-speaking country, or non-English-speaking country. Education was categorised by the highest level of attainment including bachelor's degree or higher, tertiary diploma or certificate, completed high school, or no completion of high school. Household structure was classified as couple with children, couple with no children, lone parent with child/children, lone person, or other. Labour force status included employed, unemployed, or not in the labour force. Location classified participants as living in a major city, a regional area, or in a rural area per the Australian Standard Geographical Classification 2001 [33]. Household disposable income, calculated by summing the income components for the previous financial year for all adults in the household, was equivalised using the modified OECD scale [34]. For each wave of data, nominal household income values were converted to quintiles of the Australian population distribution using percentile statistics for the corresponding financial year from the Australian Bureau of Statistics biennial Survey of Income and Housing. The lowest quintile of the population distribution of disposable household income was used as the referent group [35].

## Statistical analysis

Cox proportional hazards regression was used to model the association between providing unpaid care and incident CVD as a survival analysis with time-varying covariates. The regression model adjusted for the wave in which observations

were recorded as well as all potential confounders except for age and gender. Age was used as the timescale of the survival analysis, rather than the year of each wave observed which was instead included as a confounder, to improve exchangeability between the caregiving and control groups. Censoring occurred due to attrition; reporting having heart disease; or the end of the observation period at wave 17, whichever of these occurred first. Censoring was assumed to be noninformative. Though there was no strong evidence of an interaction effect on the association between caring and CVD by gender (p = 0.289), we elected to stratify analysis by gender due to differences in the provision of unpaid care in the Australian population between males and females [7], as well as differences in the clinical development of CVD by gender [36]. We calculated cluster-robust standard errors to account for the correlation within clusters across repeated measurements over time. We tested for heterogeneity in all variables included in the survival analyses using a global test of proportional hazards. This test assessed whether Schoenfeld residuals were correlated with time, which would indicate a violation of the proportional hazards assumption and suggest heterogeneity. No evidence of heterogeneity was found for caring intensity or any other variables for either males or females, except for long-term health conditions in females. As a result, we stratified the models by long-term health conditions for females to account for this heterogeneity in the survival analyses. Given the relatively low proportion of missing data (9.4%) and analysis of missing data (S1 Table) supported data missing at random, a complete case analysis was undertaken. All analyses were conducted using Stata version 17 [37].

## Results

Table 1 presents descriptive statistics for the study sample. The analytic sample comprised 11,123 participants, stratified by gender into two subgroups of 5,927 females (53.3%) and 5,196 males (46.7%). During the period of observation, which totalled 124,080 person-years, 686 incident cases of heart disease were reported, occurring at a rate of 55.3 cases per 10,000 person-years. The mean total follow-up time for all participants was 11.8 years. Among females, there was 326 cases of heart disease reported in 67,082 person-years under observation, an incidence rate of 48.6 cases per 10,000 person-years. The incidence rate for males was higher, at 63.2 cases of heart disease per 10,000 years, with 360 incident cases observed in 56,998 person-years. Missingness was disproportionately represented by males, those over the age of 75 years, and those living in other types of household structures (S1 Table). More females (12.4%) than males (8.6%) provided any amount of care for an elderly or disabled relative. The mean age at baseline (HILDA wave 3) was 45 years for females and 44 years for males, with most participants being between 25 and 64 years of age (73.3% and 74.6% for females and males respectively). A larger proportion of female participants in our sample identified as being an Indigenous Australian than males. There were similar levels of immigration among females and males with 22.7% of female and 23.7% of males born outside of Australia. Just under half (42.0%) of female participants and just over half (56.6%) of male participants had a tertiary qualification. Single parenting was common among female participants, with 11.9% of females living alone with children. Nearly three-quarters of male participants (74.4%) lived with a partner, regardless of whether or not they had children. Most participants (74.3%) did not have a long-term health condition. Unemployment levels were similar for both female and male participants, and a substantial proportion of females (38.9%) were not in the labour force. Approximately 62% of both females and males resided in major cities and nearly half of females (49.2%) and males (54.5%) were in the two highest population quintiles of disposable income.

Table 2 presents the results of the age-adjusted model. Findings suggested that there was an increased risk of incident CVD among females providing at least 14 hours of care per week compared with female non-carers. The estimated hazard ratio of 1.60 with a corresponding 95% confidence interval indicating that an increased risk of incident CVD for female high-intensity carers could be as little as 5% or as much as 144% more than the incident CVD risk for female non-carers. No difference was observed between females who provided care for less than 14 hours per week and non-carers. This model also estimated that there was no difference in the hazard of incident CVD for males in either caregiving category compared with males who did not provide any care.

**Table 1. Descriptive statistics of analytic sample.**

| | Female | Male | Total |
|---|---|---|---|
| Number (%) of participants | 5,927 (53.3%) | 5,196 (46.7%) | 11,123 |
| Mean (SD) follow-up time (years) | 11.94 (4.75) | 11.69 (4.84) | 11.82 (4.79) |
| Cases of heart disease reported | 326 | 360 | 686 |
| Caregiving (%) | | | |
| None | 87.6 | 91.4 | 89.4 |
| <14 hours/week | 8.8 | 6.3 | 7.7 |
| ≥14 hours/week | 3.6 | 2.3 | 3.0 |
| Baseline Characteristics at wave 3 | | | |
| Age in years Mean (SD) | 44.51 (16.69) | 43.80 (16.11) | 44.48 (16.42) |
| Age group (%) | | | |
| 18–24 | 12.5 | 13.2 | 12.8 |
| 25–34 | 18.7 | 18.7 | 18.7 |
| 35–44 | 23.1 | 23.1 | 23.1 |
| 45–54 | 18.6 | 19.3 | 18.9 |
| 55–64 | 12.9 | 13.5 | 13.2 |
| 65–74 | 8.5 | 8.4 | 8.4 |
| ≥75 | 5.7 | 3.9 | 4.9 |
| Country of birth (%) | | | |
| Australia | 75.2 | 74.9 | 75.1 |
| Other English speaking | 9.8 | 11.4 | 10.5 |
| Non-English speaking | 12.9 | 12.3 | 12.6 |
| Australia of Indigenous origin | 2.1 | 1.5 | 1.8 |
| Education (%) | | | |
| School not completed | 40.9 | 28.5 | 35.1 |
| Finished high school | 17.1 | 14.9 | 16.0 |
| Diploma/certificate | 20.7 | 36.8 | 28.3 |
| Bachelor's degree or higher | 21.3 | 19.8 | 20.6 |
| Household structure (%) | | | |
| Couple with no children | 28.4 | 30.1 | 29.2 |
| Couple with children | 39.7 | 44.3 | 41.9 |
| Lone parent with children | 11.9 | 5.2 | 8.8 |
| Lone person | 15.7 | 15.4 | 15.6 |
| Other | 4.3 | 4.9 | 4.6 |
| Long-term health condition (%) | | | |
| Yes | 25.3 | 26.2 | 25.7 |
| No | 74.7 | 73.8 | 74.3 |
| Labour force status (%) | | | |
| Employed | 58.4 | 75.2 | 66.2 |
| Unemployed | 2.8 | 3.5 | 3.1 |
| Not in the labour force | 38.9 | 21.3 | 30.7 |
| Location (%) | | | |
| Major city | 62.6 | 62.0 | 62.3 |
| Regional | 35.1 | 35.8 | 35.4 |
| Remote | 2.2 | 2.2 | 2.2 |

*(Continued)*

**Table 1.** (Continued)

| | Female | Male | Total |
|---|---|---|---|
| Disposable income (%) | | | |
| Quintile 1 (lowest) | 15.2 | 12.8 | 14.1 |
| Quintile 2 | 17.3 | 14.5 | 16.0 |
| Quintile 3 | 18.3 | 18.1 | 18.2 |
| Quintile 4 | 21.9 | 23.4 | 22.6 |
| Quintile 5 (highest) | 27.3 | 31.1 | 29.1. |

**Table 2. Age-adjusted estimates of CVD hazard ratios by caring intensity and stratified by gender.**

| | Female | | | Male | | |
|---|---|---|---|---|---|---|
| Caring intensity | HR | 95% CI | p-value | HR | 95% CI | p-value |
| No caring | ref | | | ref | | |
| < 14 hours/week | 0.78 | (0.49 - 1.23) | 0.280 | 0.85 | (0.56 - 1.30) | 0.449 |
| ≥ 14 hours/week | 1.60 | (1.05 - 2.44) | 0.027 | 0.99 | (0.57 - 1.70) | 0.957 |

The results of the primary analysis model are displayed in Table 3. This model included the wave at which observations were recorded as well as potential confounders as time-varying covariates. After adjusting for these covariates, the risk estimates were attenuated, and the confidence intervals included the null across all models.

## Discussion

In this sample of Australian adults, there was little evidence that unpaid caring is associated with increased CVD risk. In age-adjusted models, there was weak evidence that higher levels of caring were associated with increased risk of incident CVD among females. However, irrespective of the amount of time spent providing care to an elderly or disabled relative, there was no difference in the hazard of CVD for male carers in comparison with their non-caregiving control group.

These findings are consistent with both the 2013 study by Buyck et al. [25] which observed no difference in the risk of incident CHD for participants who provided care to an elderly or disabled relative and non-carers, as well as the 2018 study by Burr et al. [24] which observed no difference in incident CVD risk for participants aged over 51 years who provided care to a parent or spouse in comparison with those who did not. However, our results are inconsistent with the findings of other studies of family caregiving [27,28], including three studies which accounted for the weekly time intensity participants spent providing care to family members [21–23]. These three studies each used different thresholds for caring

**Table 3. Covariate Adjusted estimates of incident CVD hazard ratios by caring intensity and stratified by gender.**

| | Female* | | | Male** | | |
|---|---|---|---|---|---|---|
| Caring intensity | HR | 95% CI | p-value | HR | 95% CI | p-value |
| No caring | ref | | | ref | | |
| < 14 hours/week | 0.79 | (0.50 - 1.26) | 0.330 | 0.84 | (0.55 - 1.28) | 0.417 |
| ≥ 14 hours/week | 1.27 | (0.83 - 1.95) | 0.267 | 0.82 | (0.47 - 1.42) | 0.475 |

*Model adjusted for wave of observation, age, Indigenous ethnicity, country of birth, education, household structure, labour force status, location, and income, and further stratified by long term health conditions

**Model adjusted for wave of observation, age, Indigenous ethnicity, country of birth, education, household structure, long-term health condition, labour force status, location, and income

intensity to be measured at but observed an association between providing care for ≥9 hours [22], ≥ 14 hours [21], and >20 hours [23] per week and CVD.

Two prior studies also stratified their results by gender, producing contrasting findings. Burr et al. [24] observed no difference in incident CVD risk for men or women who were aged 51 years or more and provided care to their parents or spouses in comparison with non-carers. Whereas the study by Ji et al. [27] observed increases in cardiovascular-related hospitalisations among spouses of cancer patients, whom the authors had assumed required unpaid care from their spouses. Three extant studies which had demarcated unpaid care by high and low intensity [21–23], all observed associations between high-intensity unpaid care and CVD, in contrast with the results of the analysis presented in this paper. All three of these studies used study populations derived from age-restricted samples, collectively including no participants who were aged below 46 years. Importantly, the use of an age-restricted sample could influence the results of a study to observe an increased CVD risk by selecting participants who might be more likely to develop CVD due to older age or other causes. It is also possible that a reduced CVD risk may be observed if the likelihood of participants being unpaid carers has increased with the prevalence of aging spouses or relatives who require short-term care in contrast with the long-term care that might be expected for an individual with a chronic condition.

In addition to variation in sampling between these studies, other reasons for inconsistencies in findings could be owed to discrepancies in the exposure and outcome definitions. For example, our analysis did not account for death due to incident CVD or verify CVD status using medical records. Other studies which did validate this information [22,25] benefited from using a more robust exposure variable, reducing misclassification bias. Additionally, there may be differences in the availability and uptake of support and welfare services for unpaid carers in differing study locations, which could serve to relieve stress and reduce the risk of CVD development. Moreover, individual differences in the quality of caregiver/care recipient relationships and the specific demands of caring for a relative with particular needs can vary across populations and modify the stress levels of different unpaid carers and thus their likelihood of developing CVD. Furthermore, the family effect of providing unpaid care [9] could also serve to modulate how this association is observed depending on the extent to which the caregiver's wellbeing is linked with the health of their care-recipient.

The results of our study may be indicative of a healthy-caregiver effect, which has been posited by Fredman et al. [16] to explain why studies of caregiver health may observe null or protective effects for carers compared with non-carers. The healthy-caregiver effect is analogous with the healthy worker effect, suggesting that the capacity for caregivers to provide care to others is aided by their experience of a more favourable health status than might be expected in the general population. This may partly explain why no association was observed in our analysis.

Like the healthy-worker effect, healthy caregivers have been hypothesised to be able to endure higher intensity stressors without experiencing a significant health detriment [16]. Vitaliano and colleagues [19] hypothesised that exposure to chronic stress may induce CVD among caregivers. The authors conceptualised a path model which outlined a framework that attempts to explain how chronic caregiving stress can lead to CHD either directly, or indirectly by inducing distress and poor health behaviours, both of which are associated with metabolic syndrome and, in turn, can manifest as CHD [19]. The researchers postulated that this pathway could be modified by an individual's vulnerability, defined in their study as anger and hostility, as well as their personal and social resources, including socioeconomic position and social support [19].

Our analysis attempted to control for the difference between regularly and occasionally engaging in chronically stressful activity by making use of high-intensity and low-intensity caregiving categories. In separating participants like this, it would be expected that the low-intensity caregiving group would have been exposed to less stress than those in the high-intensity care group, who would be hypothesised to be more likely to develop CVD as outlined in the pathway suggested by Vitaliano et al. [19]. However, because our analysis could not account for the amount of care provided by participants in the years preceding 2003, the distinction between high- and low-intensity care only measures the frequency of caregiving over the course of one year and may not be as informative as what might be observed if caring provision was

distinguished between long-term care over multiple years and short-term care. Those who have regularly provided unpaid care to others over the course of several years would be expected to have been exposed to greater levels of stress, and thus, hypothesised to be more likely to develop CVD during the study observation period.

The findings of this study provide no evidence that intensity of unpaid care (low < 14hrs/week or high >14hrs/week) is associated with risk of developing CVD among males, and weak evidence that high intensity unpaid caring may be associated with incident CVD among females in the Australian adult population. As this is the first study of its kind to be conducted using an Australian population, these results and implications should be considered a starting point. We cannot be sure that the association observed among females is directly related to caregiving intensity, but further research is required to corroborate these findings in another larger sample. Additionally, further research may benefit from taking a broader definition of unpaid care, such that it is not delimited to caregiving for disabled or elderly relatives, but also incorporates the other types of care needs and caring relationships that occur in the population. Finally, other nuances in the association between unpaid care and CVD that warrant future exploration include: the specific relationships between caregiver and recipient including how they know each other; which activities carers assist with; and the health condition that has led the care recipient to require unpaid care; given each of these may influence the stresses experienced by an individual caregiver.

## Strengths and limitations

This study has several strengths. Firstly, the survival analysis methods to assess the effect of changes in care over time on CVD were carefully considered, with time-varying covariates selected according to a Directed Acyclic Graph as being plausible common causes of caregiving and CVD. Moreover, using age as the timescale in our survival analysis model allowed for comparisons of CVD risk to be made among participants of the same age, rather than participants of disparate ages, who were observed at the same point in time. Although left truncation is associated with using age as a timescale in survival analysis, causing a survivorship biasing effect, this is unlikely to affect our model as the onset of CVD is unlikely to occur in participants younger than the ages captured in our sample. This study also had the advantage of using data drawn from a large, nationally representative survey which should improve the external validity of this study and thus improve the generalisability of these results into the wider Australian population. Lastly, in addition to contributing to the small number of studies examining intensity of care provision, this is the first study to our knowledge to examine the association between unpaid care provision and CVD in an Australian population.

This study also had some limitations. First, generalisability of these findings to the broader Australian population may be limited since the HILDA survey under-represents people of lower socioeconomic positions, as well as migrants/new Australians [38]. As all variables were self-reported, there is some risk of self-reporting bias. Given that some carers may not identify as carers, the self-reported nature of the collected variables could lead to bias due to exposure misclassification, potentially underestimating associations between caregiving and CVD. Furthermore, the measure of caregiving available in this data does not capture variations in the intensity and duration, nor type of caregiving. Notably, the caregiving measure did not include those who provided caregiving for someone who is neither elderly nor has a disability, potentially underestimating true associations. Non-familial caregiver/care recipient relationships are also likely excluded by the definition of the exposure variable that was used, which could misclassify caregiving status and cause an underestimation of associations. Misclassification of the outcome, which was measured as self-reported CVD status, may also have occurred whereby participants may have not reported having CVD despite having CVD – either because they were unaware or because they did not want to disclose this information – potentially resulting bias toward the null.

There is also potential for bias due to missing data, as although the final analytic sample only contained participants who had complete data for exposure, outcome, and covariates, there was a small proportion of eligible participants with missing data who were therefore excluded from the analysis. The demographic differences at baseline between those who were removed from the study population due to missing data and those included in the analytic sample are included

as supplementary material (S1 Table). It is notable that among participants with missing data, there were more males, more individuals over 75 years of age, and more participants in other household structure types than the set of participants which comprised the sample used for analysis. Given that these factors might be associated with increased CVD risk, it is possible that this analysis may have underestimated the risk of CVD compared with what would be observed if complete data for baseline CVD status was available for all eligible participants.

Furthermore, this study may have been limited by a lack of statistical power due to the small number of incident CVD events observed during the follow up period. This lack of power may have been exacerbated in our gender stratified analysis, potentially leading to imprecision in our risk estimates. Lastly, the extent to which these results may be generalisable to populations of carers in other countries is unclear. Acknowledging the limitations of this study in addressing the research question, it is clear that further research is needed to interrogate the relationship between caregiving and CVD, with more inclusive measures of care provision and accounting for types of caregiving required (i.e., what activities require assistance, is the caregiving long-term or short-term?) and also the nature of the caregiver/care-recipient relationship (familial or otherwise).

## Conclusion

This study addressed key research gaps in the literature examining the effects of providing unpaid care on cardiovascular health. To our knowledge, this is the first such study to be conducted within the Australian population. In addition to this, this study contributes to the small number of studies that have described the difference in CVD risk between low and high intensity caregivers. In age- adjusted models, there was weak evidence of an association between unpaid caring intensity and incident CVD among females, but no association was observed for females in fully adjusted models, nor was there any increase in the risk of incident CVD for male Australian adults providing unpaid care compared with those who did not. These findings establish a base understanding of how caregiving may influence the cardiovascular health of Australian adults and highlight a need for further research to more clearly understand this association, both in Australia and globally.

## Supporting information

**S1 Table. Baseline (wave 3) characteristics of participants with missing data and no missing data.**
(DOCX)

## Author contributions

**Conceptualization:** Ameer Lambrias, Yamna Taouk, Jennifer Ervin, Tania King.

**Data curation:** Ameer Lambrias, Yamna Taouk, Jennifer Ervin.

**Formal analysis:** Ameer Lambrias, Yamna Taouk.

**Funding acquisition:** Tania King.

**Investigation:** Yamna Taouk, Jennifer Ervin, Tania King.

**Methodology:** Ameer Lambrias, Yamna Taouk, Jennifer Ervin, Tania King.

**Resources:** Tania King.

**Software:** Ameer Lambrias, Yamna Taouk.

**Supervision:** Yamna Taouk, Jennifer Ervin, Tania King.

**Validation:** Ameer Lambrias.

**Visualization:** Ameer Lambrias.

**Writing – original draft:** Ameer Lambrias.

**Writing – review & editing:** Ameer Lambrias, Yamna Taouk, Jennifer Ervin, Tania King.

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
