## [Decision Letter · Decision Letter 0]

4 Jun 2024

Dear Dr. Lambrias,

Thank you for submitting your manuscript to PLOS ONE. After careful consideration, we feel that it has merit but does not fully meet PLOS ONE’s publication criteria as it currently stands. Therefore, we invite you to submit a revised version of the manuscript that addresses the points raised during the review process.

We only managed to get one reviewer to review.

Please make the changes as needed. 

We look forward to receiving your revised manuscript.

Kind regards,

Yee Gary Ang, MBBS MPH

Academic Editor

PLOS ONE

3. In the online submission form you indicate that your data is not available for proprietary reasons and have provided a contact point for accessing this data. Please note that your current contact point is a co-author on this manuscript. According to our Data Policy, the contact point must not be an author on the manuscript and must be an institutional contact, ideally not an individual. Please revise your data statement to a non-author institutional point of contact, such as a data access or ethics committee, and send this to us via return email. Please also include contact information for the third party organization, and please include the full citation of where the data can be found.

Reviewers' comments:

Reviewer's Responses to Questions

**Comments to the Author**

1. Is the manuscript technically sound, and do the data support the conclusions?

Reviewer #1: Yes

2. Has the statistical analysis been performed appropriately and rigorously?

Reviewer #1: I Don't Know

3. Have the authors made all data underlying the findings in their manuscript fully available?

Reviewer #1: No

4. Is the manuscript presented in an intelligible fashion and written in standard English?

Reviewer #1: Yes

Reviewer #1: The paper is of reasonable quality. However, there are major changes to how the analysis is conducted and written up that have to change before it is ready for publication. I am recommending these points are comprehensively addressed before proceeding to publication.

-- Major points --

## Analysis considerations ##

- Outcome measure - include the exact wording of the question. It's hard to understand the outcome in this paper without you detailing what the question is. Furthermore, are you trying to determine something equivalent to incident CVD? Is that why people with CVD at baseline were excluded.

- Population - how many people in the younger age groups reported they had heart disease? If it is very few, I think these people should be excluded from the analysis and you should focus on the age bands where heart disease starts to emerge. I know you discuss this later in the study, but it's an important consideration in the calculation of your outcome. Including younger people increases the size of the denominator, which lowers the risk of CVD, which could impact the estimation of relative inequalities.

- Relative and absolute inequalities need to be presented. this could help explain why women tend to have higher relative inequalities (in the unadjusted model at least) as the baseline level of CVD among female non-carers is lower.

- Confounders - I don't understand why you haven't included a table for the confounders split by the exposure. I have no way of knowing what was happening in your adjustment.

- Time varying confounders - I think this is tricky, it just feels like the variables have been dumped into the model without thinking through the causal structure. Is income a confounder? It can lead to someone in the household acquiring disability or care needs. Caring can then lead to pressure on ability to work, potentially lowering income. In this case it switches from being a confounder at time point 1 to being a mediator at time point 2. The DAG would be something like: Income -> Caring -> Income -> CVD. I think this could materially affect your results given adjustment seems to knock out a lot of the effect for women. If caring -> income, you have just conditioned on a mediator, and you are masking the impact of caring -> CVD.

- Coding of income - maybe less important, and I don't have a strong view, but thought needs to be given as to whether household income is the best measure.

- Results, statistical interpretation - this is currently wrong. You haven't found there is "no difference", you just don't have sufficient evidence to reject the null hypothesis.

- Table 1 - I don't understand why there are so few observations in the 18-24 group. Is this because they didn't answer the CVD question - hence were excluded from the study.

- Limitations section - I think the limitations section needs to be much clearer. How does a lack of statistical power lead to underestimation of risk? It leads to lower precision. If the smaller sample size is a result of missing data that affects some groups more than others, or attrition related to the outcome or exposure, then that could lead to bias. But a small sample on its own does not necessarily lead to bias, just imprecise estimates. I think each of the separate forms of bias discussed need to be set out clearly, with an assessment of which direction they could bias the results. This could even be done using bullet points or one bias consideration per paragraph. Currently it feels a little like word soup.

- Strengths - I am worried that what you think is a strength - how you've chucked time varying measures into the model, is in fact a weakness and is risking biasing your results. Please think this through carefully.

-- Minor points --

- Have you checked how your incidence / rates of CVD compare to what is observed in other datasets? This will help you understand what you have captured in your data. Potentially use the Census to check - I think there are chronic health condition questions in the census now, the same question might have even been used!

**Do you want your identity to be public for this peer review?** For information about this choice, including consent withdrawal, please see our Privacy Policy

Reviewer #1: No

---

## [Author Response · Author response to Decision Letter 1]

23 Oct 2024

Manuscript reference number: PONE-D-23-41861

Cardiovascular Disease Risk Among Australian Informal Carers – A Survival Analysis using 15 waves of the HILDA Survey

We thank the reviewer and editors for their time and considered comments and believe that in addressing them, we have substantially improved our manuscript. Our specific responses to each of the comments are detailed in our Response to Reviewers document attached.

---

## [Decision Letter · Decision Letter 1]

13 Dec 2024

Dear Dr. King,

Thank you for submitting your manuscript to PLOS ONE. After careful consideration, we feel that it has merit but does not fully meet PLOS ONE’s publication criteria as it currently stands. Therefore, we invite you to submit a revised version of the manuscript that addresses the points raised during the review process.

One of the reviewers have recommended rejection but the other reviewer has suggested accept 

We invite you to revise extensively before we invite the reviewers to review again

We look forward to receiving your revised manuscript.

Kind regards,

Yee Gary Ang, MBBS MPH

Academic Editor

PLOS ONE

Reviewers' comments:

Reviewer's Responses to Questions

**Comments to the Author**

Reviewer #2: All comments have been addressed

Reviewer #3: (No Response)

2. Is the manuscript technically sound, and do the data support the conclusions?

Reviewer #2: Yes

Reviewer #3: Partly

3. Has the statistical analysis been performed appropriately and rigorously?

Reviewer #2: Yes

Reviewer #3: No

4. Have the authors made all data underlying the findings in their manuscript fully available?

Reviewer #2: No

Reviewer #3: No

5. Is the manuscript presented in an intelligible fashion and written in standard English?

Reviewer #2: Yes

Reviewer #3: Yes

Reviewer #2: The manuscript focuses on important questions and the risk of developing CVD among informal caregivers in the Australian context is under studied. The paper is well written and I believe the authors have comprehensively addressed the detailed comments and requests from the prior review. It is notable that the HILDA dataset used in the current analysis contained a relatively large amount of missing data on CVD status among participants who were older, informal caregivers, indigenous Australians from lower income brackets and the limitation is rightly described. Further studies are therefore required to validate these findings.

Reviewer #3: This review evaluates the manuscript titled “Cardiovascular Disease Risk Among Australian Informal Carers – A Survival Analysis using 15 waves of the HILDA Survey”. The study explores the relationship between informal caregiving and the risk of cardiovascular disease (CVD), given the potential stress associated with caregiving. However, the authors did not find any evidence suggesting an increased risk of CVD due to informal care.

Major Comment:

The research topic is engaging and relevant; however, the study’s findings do not support the hypothesized link between stress from caregiving and CVD risk. My primary concern lies in the study design, as it appears the analysis was tailored to the research question, rather than the research question driving the analysis.

The outcome variable, CVD, reflects a heterogeneous population. For instance, the inclusion of participants as young as 18 years old introduces variability, particularly since the youngest participants in the later waves are only 33 years old, an age group with a relatively low likelihood of CVD incidence. Additionally, informal caregiving intensity and associated stress levels vary widely based on individual circumstances and the nature of care, further contributing to data heterogeneity. This high heterogeneity likely increases variance, which can make it difficult to reject the null hypothesis.

Although the authors conducted sensitivity analyses on limited subpopulations, they employed the same methodology without adequately addressing the heterogeneity. Before conducting specific analyses such as survival analysis, I would recommend basic statistical investigations to better understand the data. For example:

• Perform bivariate analyses to examine relationships between age groups, CVD incidence, informal caregiving intensity, and stress levels.

• Investigate high-risk groups, such as middle-aged single mothers with caregiving responsibilities and high K10 scores, to determine their CVD rates.

• Use crosstab analyses to assess CVD rates among subgroups with informal caregiving responsibilities and varying stress levels.

• Validate these findings using general logistic regression before progressing to survival analysis, ensuring the analysis is warranted and appropriately contextualized.

By addressing these foundational issues, the study could more robustly address the research question and manage the heterogeneity in the data.

Minor Comments:

1. There is inconsistency in how the waves and years of data are described. My understanding is that the authors used waves 3, 7, 9, 13, and 17. However, the manuscript implies data spanning waves 3–17. This should be clarified and corrected.

2. The analysis appears incomplete. Additional analyses addressing the heterogeneity and variability in the dataset are necessary to strengthen the study. A missing data analysis is relevant.

3. The literature review should be updated to include recent research on Australian informal care, particularly studies focusing on care for disabled or elderly individuals.

**Do you want your identity to be public for this peer review?** For information about this choice, including consent withdrawal, please see our Privacy Policy

Reviewer #2: No

Reviewer #3: No

---

## [Author Response · Author response to Decision Letter 2]

9 Feb 2025

NOTE: These have been attached with other submission documents. Our response contained tables and the formatting of these did not work when pasted in this box.

---

## [Decision Letter · Decision Letter 2]

6 Apr 2025

Cardiovascular Disease Risk Among Australian Unpaid Carers – A Survival Analysis using 15 waves of the HILDA Survey

PONE-D-23-41861R2

Dear Dr. King,

We’re pleased to inform you that your manuscript has been judged scientifically suitable for publication and will be formally accepted for publication once it meets all outstanding technical requirements.

Kind regards,

Yee Gary Ang, MBBS MPH

Academic Editor

PLOS ONE

Additional Editor Comments (optional):

Reviewers' comments:

Reviewer's Responses to Questions

**Comments to the Author**

Reviewer #4: All comments have been addressed

2. Is the manuscript technically sound, and do the data support the conclusions?

Reviewer #4: Yes

3. Has the statistical analysis been performed appropriately and rigorously?

Reviewer #4: Yes

4. Have the authors made all data underlying the findings in their manuscript fully available?

Reviewer #4: Yes

5. Is the manuscript presented in an intelligible fashion and written in standard English?

Reviewer #4: Yes

Reviewer #4: I have gone through both rounds of revisions, and I am satisfied with the authors' revisions. I think the final draft is of much better quality. I have no more comments.

**Do you want your identity to be public for this peer review?** For information about this choice, including consent withdrawal, please see our Privacy Policy

Reviewer #4: No

---

## [Editor Report · Acceptance letter]

PONE-D-23-41861R2

PLOS ONE

Dear Dr. King,

I'm pleased to inform you that your manuscript has been deemed suitable for publication in PLOS ONE. Congratulations! Your manuscript is now being handed over to our production team.

Kind regards,

on behalf of

Dr. Yee Gary Ang

Academic Editor

PLOS ONE